# Understanding Motivations for Individual and Collective Sustainable Food Consumption: A Case Study of the Galician Conscious and Responsible Consumption Network

Isabel Lema-Blanco [1,*], Ricardo García-Mira [1] and Jesús-Miguel Muñoz-Cantero [2]

1   People-Environment Research Group, Faculty of Educational Sciences, University of A Coruña, 15071 A Coruña, Spain
2   Department of Specific Didactics and Methods of Research and Diagnosis in Education, Faculty of Educational Sciences, University of A Coruña, 15071 A Coruña, Spain
*   Correspondence: isabel.lema@udc.es

**Abstract:** Although consumer attitude towards sustainable food has increased over time, scientific research suggests that more profound comprehension is needed of the social and psychological dimensions that condition consumers' willingness to purchase food items produced in a sustainable way. The current study attempts to understand the individual motivations that drive conscious consumption, in both the individual and collective spheres, expressed through joining a local social innovation in the food domain. A multi-method design was used, which combined participatory observation, document analysis, and twenty-six in-depth interviews with members of eight local food consumption initiatives located in the Galician region (Spain). The findings reveal that sustainable food purchasing is driven by the individual's motivation to maintain a healthy diet, pro-environmental and social values, environmental awareness, and attachment to Galician rural areas. Concerning motivations underlying people's willingness to engage in conscious and responsible consumption initiatives, the first reason relates to the fulfilment of basic needs (affordability and accessibility to organic, low-carbon, and fair-trade goods), followed by sociopolitical goals and the aspiration to satisfying social and psychological needs such as the need for connectedness or the desire for autonomy and control over one's purchasing decisions.

**Keywords:** sustainable food consumption; motivations; sustainable local innovations; food movement

---





## 1. Introduction

The Sustainable Development Goals (SDGs) adopted by the United Nations [1] require relevant societal changes that involve the development and support of innovative approaches to protect Earth's ecosystems and fight against ecological degradation and the loss of biodiversity. Among the 17 SDGs included in the 2030 Agenda, the 12th goal focuses on sustainable consumption and production patterns, recognizing that the global environmental crisis is driven by people's resource-intensive lifestyles, social needs, and aspirations [2]. Food systems have been indicated as one of the greatest drivers of poor health and environmental degradation globally [3,4]. For instance, the food sector accounts for around 30% of the world's total energy consumption and around 22% of total greenhouse gas emissions [5]. Some voices [6] have also been critical concerning the voluntary and indirect policy approaches for achieving sustainable consumption and production patterns. Reaching this goal involves transitioning towards low-carbon and circular economy approaches in food systems as well as expanding the wide adoption of sustainable lifestyles in Western societies [7,8].

Social scientists have long been concerned with the challenge of empowering people to lead the ecological transition, which involves the adoption of sustainable patterns of

consumption [7,8]. Previous literature has mostly focused on individual and consumer-centric understanding of sustainable consumption. However, this perspective seems limited, as consumer empowerment dynamics should also be analyzed in relation to the collective creation of alternative modes of social organization in the economy, e.g., citizen-led grassroots innovations [9], which are the main focus of the current study.

This article consists of five sections. The second section focuses on the conceptual and theoretical framework of sustainable consumption, with a specific focus on factors underlying conscious and responsible consumption as well as motivations for people to engage in grassroots innovations in the food domain. Section 3 contextualizes the study and provides a detailed description of the research objectives, methodology, and data sources used in the article. Section 4 presents the main results on the underlying motivations for organic and sustainable food consumption and for joining a local conscious and responsible consumption initiative. Finally, Section 5 opens the discussion and presents the main conclusion of the study.

## 2. Conceptual and Theoretical Framework

### 2.1. Sustainable Consumption

Sustainable consumption was defined at the Oslo Symposium [10] as "the use of services and related products, which responds to basic needs and bring a better quality of life while minimizing the use of natural resources and toxic materials as well as the emissions of waste and pollutants over the life cycle of the service or product so as not to jeopardize the needs of further generations". Sustainable consumption relates to habits and patterns of behavior that are embedded in society, and fostered or inhibited by institutions, norms, and infrastructures, that frame people's choices [11]. Citizens, as consumers, need the capacity and the motivation to make decisions about their consumption patterns and choose available low-carbon options, change what they eat and purchase, and organize or participate in community responses to climate change [7,8]. Citizens are said to consume sustainably for its symbolic, rather than its economic value, as goods play vital symbolic roles in people's lives, and are used as instruments to communicate, for example, their status or identity [12–15]. Further, motivations are context-dependent and are conditioned by cultural, organizational, and societal features that mediate people's willingness to endorse green alternatives [1,15–17]. For example, people's energy-related decisions are characterized by the satisfaction of basic human needs and aspirations that contribute to the endorsement of sustainable social innovations and energy transitions [18–20]. However, as several scholars have stressed [7,8,21], more profound knowledge is needed about the influence of psychological aspects in driving citizens' motivations to engage in sustainable consumption in different domains such as energy use, food intake, or mobility.

### 2.2. Factors Influencing Sustainable Dietary Choices

Food choices have great significance in terms of mitigating climate change. According to recent IPCC reports, extensive meat production accounts for 18% of global GHG emissions, an impact that is increased by emissions derived from its transportation and distribution. However, in spite of increasing environmental knowledge and global warming awareness, a notable gap between people's positive attitudes towards sustainable and organic food and their dietary routines has been reported [14,21]. Eating habits are notoriously difficult to change, as they are deeply culturally and structurally embedded and influenced by a wide range of social, cognitive, socioeconomic, and contextual factors. For example, organic food intake seems to be strongly influenced by educational level, gender, age, and level of income. In terms of gender and age, women and young people are more likely to buy organic food [22,23]. Moreover, many studies showed that the price and affordability of organic products in the market strongly influence consumers' choices, and higher prices are argued as one of the most important barriers to purchase organic food [24,25]. Concerning education, research has found that educated people (e.g., university degree level) usually present greater preferences for organic food [24,25].

Another stream of research has focused on the influence of personal characteristics such as values and beliefs or identities. Psychological literature stresses that people who reveal altruistic or prosocial values are more likely to maintain more responsible behavior with respect to the environment [8,11,26]. A few studies have also found that vegetarianism can be positively related to altruistic values and negatively to traditional values [26], while biospheric values underly the purchase of organic food or ethical and fair-trade products [14,27]. Furthermore, people attribute a series of symbolic or affective connotations to certain goods. There is also empirical evidence that suggests that vegetarianism and meat consumption are closely related to people's self-concept or identity [12]. Further, apart from health, environmental awareness, and animal rights, dietary choices and eating behaviors are strongly influenced by the social groups people belong to and existing social norms, for example, on meat consumption or vegetarian choices [28].

Recent studies also point to the existence of specific factors affecting organic food choices such as health orientation, sensory appealing, and functionally characteristics. Health orientation relates to individuals' motivations to sustain healthy lifestyles. There is extensive empirical evidence showing that health concerns are particularly relevant drivers for organic food consumption [14,28] or lower meat consumption [29]. Thus, organic groceries are perceived as healthier as they are not exposed to harmful chemicals, are free of GMOs, and have higher quality and superior flavor of the product.

Research also found that sensory appeal and functionality characteristics are influential factors for buying organic products and foods. According to recent studies, hedonistic benefits such as sensory characteristics can stimulate the purchasing of organic groceries [24,25]. For example, a study conducted by Chekima et al. [24] found that consumers of organic products reported a strong influence of sensory attractiveness. People are stimulated by the sensory characteristics (e.g., taste), which means "a consequence of the functional and psychological benefits provided by the product and exerts its effect on the choice of food through the negotiation of values by the consumer" (ibid, p. 1445). These authors point out that although the predominant approach in organic products marketing strategies stresses the health or environmental benefits, an emphasis on the hedonic aspects of these foods could be a novel effective approach.

A number of marketing-oriented studies have used the means–end chain theory [30] to explore values-based motivations underlying sustainable and organic food purchasing decisions. This theory states that products' attributes are means for consumers to obtain desired ends. It connects the tangible attributes of a product (the means) to highly abstract and intangible personal and emotional values (the ends). According to these scholars [31,32], consumers associate organic products with health benefits; however, they also share desire for high-quality, tasty and nourishing food, as wellbeing and pleasure appear as the most relevant values.

### 2.3. Motivations and Aspirations Underlying Collective Forms of Conscious and Socially Responsible Food Consumption

A parallel line of research explored intrinsic motivations and aspirations underlying ethical consumption as well as people's willingness to engage in new social movements and grassroots innovations in the food domain. These social innovations usually operate in civil society arenas and involve networks of activists and organizations generating novel solutions for sustainable development [33,34]. Conscious and responsible consumption relates to an individual's awareness of the social and ecological impact of his/her consumption choices and, consequently, the decision to avoid products or services from companies that they perceive as harmful to both society and the environment and the preference for products or services from companies that benefit society [34].

Recent studies on grassroots local innovations in the food domain show that the perceived high-quality of organic products, environmental concern, and the desire to strengthen the local economy represent the main motivations for people to purchase organic, "kilometer zero" food [35–37]. The activists of these grassroots innovations are

also attracted by the societal benefits derived from these local food networks such as the democratization of the food system, the conservation of the food culture and traditions, and the contribution to social cohesion in urban contexts [36,37].

Research on local organic agri-food networks presented evidence that conscious consumers are guided by altruistic motivations, social values, attachment to the territory, or the search for more supportive, ethical, and sustainable production food-systems [34–38]. Further, community activities related to food can stimulate the satisfaction of intrinsic motivations, such as the aspiration to be more autonomous in accessing food products and less dependent on external actors. For example, in a study on the Slow Food movement, Dumitru and Lema-Blanco et al. [37] found that the opportunity to connect with other like-minded people was the main attraction for many activists, since interaction with other people reinforces their sense of connection and their experience of happiness and self-esteem, which represents a psychological reward for participating in these initiatives.

## 3. Materials and Methods

### 3.1. Case Study Description

This study is contextualized in the Galician region (Spain). Specifically, the consumption initiatives that are part of the so-called "conscious and responsible consumption network" were studied as a manifestation of citizen engagement in the Galician food movement. This informal network, which is formed by twenty-five collective food buying groups and cooperatives, engages a total of 1500 families across the region (see Figure 1, below). These citizen-led initiatives are formed by consumers that collectively organize their food purchases by establishing agreements with farmers and local producers, with whom they establish regular deliveries of seasonal products [38,39]. Adopting different organizational and legal formulas (food coops, non-profit associations, and informal consumption groups), the Galician conscious and responsible consumption initiatives claim to promote the consumption of organic and locally produced food, as well as other fair trade and solidarity products. They usually share common principles and values aligned to the social and solidarity economy and food sovereignty movements. These local initiatives define themselves as transformative projects that aim to build, at the local level, a more egalitarian, fair, sustainable, and more democratic food system [38,39].

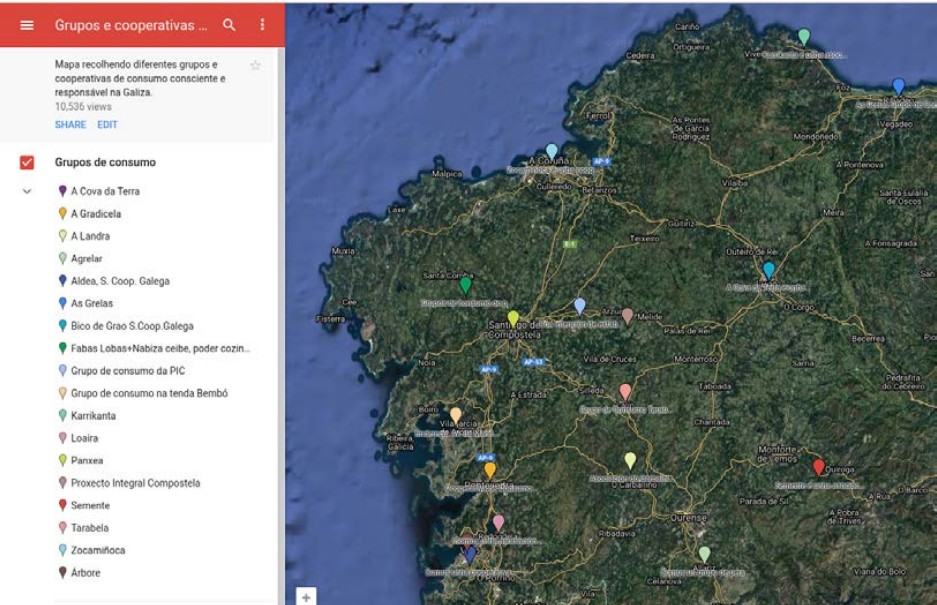

**Figure 1.** Collaborative map of the Galician conscious and responsible consumption network that illustrates the distribution of the local initiatives across the Galician territory. Source: Google Maps.

*3.2. Research Objectives, Methods, and Sample Description*

Focusing on the need to increase understanding about the social and psychological dimensions driving sustainable consumption, the overall goal of this research is to carry out a holistic study on the role of the Galician food movement in the promotion of sustainable lifestyles and climate-responsible behaviors. Based on the theoretical review carried out, the following research questions were defined:

1. What are the social and psychosocial factors that influence sustainable food choices within the Galician food activists?

2. What are the motivations underlying citizen's engagement in sustainability initiatives, specifically conscious and responsible consumption initiatives?

A qualitative interpretative approach [40,41] was adopted to conduct and holistic and inductive-based research on the complex and dynamic phenomena of the conscious and responsible consumption network. A multi-method design was used combining participatory observation, document analysis, and twenty-six in-depth interviews with participants in eight local consumers' initiatives located across the Galician region (Spain).

The empirical design was structured in four phases: phase I consisted of a desktop documentary review of the Galician food initiatives; phase II consisted of participant observation that extended throughout the entire empirical phase of the study; phase III consisted of exploratory interviews with a sample of fourteen activists of three local initiatives located in the cities of Vigo (Árbore, Aldea) and A Coruña (Zocamiñoca) (data collection took place in the second half of 2016 and early 2017; in phase IV, twelve semi-structured interviews were conducted with current and former members of seven consumption initiatives (A Gradicela, Agrelar, Árbore, Millo Miúdo, Panxea, Semente, and Zocamiñoca). Data were collected between the months of November 2017 and August 2018.

The interviews were conducted following an interview guide, raising the topics in an open and flexible way, with the aim of allowing new questions to emerge and for the interviewees to freely express their subjective experiences, opinions, and reflections regarding the suggested topics. The topics covered in the empirical study revolved around food activist' lifestyles, motivations for conscious and responsible consumption, organizational features and social learning processes rising within these consumers' initiatives.

Table 1, below, illustrates the distribution of the sample in terms of the food initiative they belong to or belonged to in the past.

**Table 1.** Description of the sample of participants according to the conscious and responsible consumption initiative to which they belong.

| Typology | Name of the Initiative | Number of Participants |
|---|---|---|
| Cooperative with a store open to the public | Árbore (Vigo, Pontevedra) | 3 |
| | Aldea (Vigo, Pontevedra) | 3 |
| Cooperative with a store (only for associates) | Zocamiñoca (A Coruña) | 11 |
| Non-profit organization | Semente (Ourense) | 2 |
| | Millo Miúdo (Oleiros, Coruña) | 4 |
| Consumers' group | Agrelar (Allariz, Ourense) | 1 |
| | A Gradicela (Pontevedra) | 1 |

Total number of people interviewed in phase III and IV: 26; total number of entities: 8. Source: own study.

A total of twenty-six participants were interviewed in the study. The sample comprised fourteen men and twelve women, aged between 35 and 65. Of the twenty-six participants, ten were aged between 35–39 years, eight were aged between 40–45 years old, five were between 46–50 years old, and three were between 51 and 65 years old. Regarding their educational level, participants shared a high educational level. Thus, sixteen of them had university master's, graduate, or postgraduate studies and five of them had a doctorate. Four participants had completed technical and vocational training studies or other specializations.

*3.3. Data Analysis Procedure*

All interviews were audio-recorded and later literally transcribed. A rigorous coding process and data analysis and interpretation were conducted supported by Atlas.ti V.9 software [42]. Data analysis followed an inductive procedure, in which the relevant themes "emerged from the data" as a result of a process of codification and categorization [43]. The analysis and constant systematic comparison of data allowed us to identify the similarities and differences and recurrent patterns, giving rise to the formulation of analytical categories. A process of relation and comparative analysis was followed among the distinct groups of codes, phenomena, or incidents, leading to the formulation of results [40].

Although the qualitative paradigm gives predominance to the study of the essential qualities of the social phenomenon, certain quantification or counting processes were conducted that allowed us to evaluate the degree of relevance of a theme or pattern and its consistency, considering the frequency of essential properties [44]. Likewise, the use of Atlas.ti software [42] allowed us to determine the relevance of the findings, providing data about the groundedness (number of citations of each code), the density (number of relationships with other codes), and the frequency (number of interviews that contain the code) of each code and category.

Informed consent was obtained from each participant, who were informed about the objectives of the research and the ethical code applied to qualitative studies. Further, in order to guarantee the confidentiality and anonymity of the participants, their names were changed in the quotations used to illustrate the results of the study.

## 4. Results

The findings of the study are presented as follows: first, the reasons and motivations driving organic and sustainable food consumption within the members of the Galician food movement are presented in Section 4.1. Following, Section 4.2 delves into the specific motivations and expectations for people who become members of a conscious and consumption initiative, which usually involves a formal commitment. The participants did not practice sustainable consumption exclusively but mostly showed a preference for organic products and claim to maintain climate-friendly behaviors in their households.

*4.1. Underlying Motivations to Organic and Sustainable Food Consumption*

When asked about their reasons to purchase organic fresh food and other products, the participants in this study argued that their decisions were motivated by a combination of factors that have been grouped into main five categories: (i) environmental concern; (ii) personal health and wellbeing; (iii) awareness-rising; (iv) altruistic and social justice values; and (v) attachment to rural areas. Table 2, below, describes the main characteristic of each category and its relevance, considering the groundedness and frequency in which each of them were referred in interviews.

**Table 2.** Motivations underlying sustainable, conscious, and responsible consumption.

| Category | Description | Groundedness and Frequency |
|---|---|---|
| Environmental concern | Environmental concern about environmental risks, climate emergency, and impact of food system. | G: 41; F: 17 |
| Health and personal wellbeing | Desire to consume healthy and organic food (free of pesticides or GMOs) for personal and your family acquires a priority role. | G: 30; F: 17 |
| Altruism and socially oriented values | Altruism concerns, desire to improve the lives of those who are in disadvantaged situations. Socially oriented values are aligned with solidarity and social justice, building new types of relations between both the Global North and South. | G: 19; F: 9 |

**Table 2.** *Cont.*

| Category | Description | Groundedness and Frequency |
|---|---|---|
| Attachment to rural | Sense of attachment to rural areas dedicated to primary sector of economy; and the desire to dignify small farmers/producers and sustain (Galician) traditional lifestyles. | G: 9; D: 2; F: 5 |
| Awareness raising | Awareness-rising due to an event or personal experience that acts as a trigger for a change in individual consumption styles and that understands eating as an essential part of a set of desirable pro-environmental behaviors. | G: 24; F: 9 |

G = groundedness, number of citations or quotations that contain a code/category. F = frequency, number of interviews that contain a code/category. Source: own study.

### 4.1.1. Environmental Concern

The category "Environmental concern" refers to participant's preoccupation about environmental risks, climate emergency, or the ecological impact of the food system. Most of the interviewees were concerned about the depletion of resources caused by the current economic model and contamination derived from the food systems; it was mentioned as one of the reasons underlying organic food choices. Environmental concern was manifested also as the interest in developing more responsible individual behavior in the private life domain, aimed at reducing the ecological footprint and minimizing resource use in households to contribute to global environmental wellbeing.

In addition, a substantial number of interviewees have been members or supporters to environmental associations (e.g., Adega, Verdegaia, Greenpeace, Ecologistas en Acción). They showed a high interest in being informed on environmental issues. In particular, they were concerned about the impact of the current globalized model of exploitation of natural resources and its effects on climate change. A critical vision was shared among the activists concerning the "unsustainable global food system" in opposition to organic and local food production practices that organic food supports, as the following quote illustrates:

*"(Organic food relates to) supporting local markets, local producers, local products, not moving food from one place to another on the planet. It is ridiculous! We want to support organic food, but within a sustainable social frame. Not the organic food that is sold in Carrefour or whatever. I mean responsibility in the entire system" (quote: Fátima, Millo Miúdo).*

### 4.1.2. Health and Personal Wellbeing

The members of the Galician sustainable consumption initiatives shared a growing concern about the impact of environmental pollution on health, which turns into a desire to eat healthier food, free of pollutants (e.g., pesticides), or not genetically modified. One of the founding members of Zocamiñoca described the profile of the members of this cooperative as follows:

*"Zocamiñoca's partner profile is a person who decides to be join Zocamiñoca for more reasons than only food needs. They are people who care about their health and who like to eat healthy. They are usually active people, which maintain a less sedentary lifestyles than regular people" (quote: Lois, Zocamiñoca).*

Organic fresh vegetables and fruits were usually reported as healthier than processed food, which may also lead the adoption of vegetarian diets. Further, several respondents mentioned that past food crises appear to have influenced citizens' awareness of food safety; for example, the "cow disease" (bovine spongiform encephalopathy) crisis that strongly affected the region in 2001 was used by veteran food initiatives such as Árbore to direct the attention of the citizenry towards organic food.

### 4.1.3. Altruistic and Socially Oriented Values

The participants in this study shared a set of altruistic and socially oriented values, aligned with solidarity and social justice, as manifested in the endorsement of new social movements such as pacifist, environmentalist, or the anti-globalization movements that call for the transformation of power relations between the Global North and South. Consumption was conceived as a powerful tool to promote structural and systemic transformation processes, confronting the capitalist system—which they describe as unsustainable and unfair—and supporting alternative models aligned with their personal values, as illustrated by the following quotation:

*"Social change is possible step by step, working every day. Not saying "we are going to make the revolution" and then you find that nothing has changed, which is what has happened throughout history. Indeed, there have been no structural changes, because we have never challenged the core parts of the system, just only the most superficial ones. Consumption is the basis. What we eat is what we are. What we walk, what we breathe, what we are and what we do "(quote: Fátima, Millo Miúdo).*

### 4.1.4. Attachment to Rural

One motivation shared by a relevant number of participants is the desire to contribute to the preservation of the Galician rural environment and the traditional lifestyles of people dedicated to the primary sector and the sustainable exploitation of natural resources. These activists were concerned about the abandonment of/loss of population in rural areas of Galicia. They were also aware of the difficulties that small farmers suffer to sustain their lifestyle, intricately linked to the exploitation of the field and the forest. Such rural attachment was often driven by their family roots. For example, several interviewees acknowledged that their family, their parents, or grandparents, or even they themselves used to live in a rural area and thus they feel attached to it, even if they do not reside there anymore:

*"I have always been in contact with the rural, although I live in the city. I have always liked it, I always have appreciated, I have always felt comfortable, and it has seemed like a place to preserve" (quote: Fabio, Zocamiñoca).*

*"I always valued the issue of food, and I always gave a lot of value to the people who work in agriculture, who are actually feeding us. In fact, after working in a garden, knowing what it is, planting your food . . . for me it was particularly important the dignity of the people who are working in the field. It was something highly relevant in the project" (quote: Xulia, Zocamiñoca).*

### 4.1.5. Awareness Raising

The category "Awareness raising" relates to a specific event or personal experience that occurred to them or happened at a specific time and that acted as a trigger for a change in individual consumption styles. Sharing pro-environmental or social values or environmental concern are relevant dimensions but are not sufficient for driving desirable changes in patterns of consumption or eating habits. Some participants delved into the interviews about their process of personal change, identifying critical situations or personal life events that lead to "awareness-raising" processes that triggered changes in their consumption habits and that were described as mechanisms of activation for their environmental identity, as the following quotation illustrates:

*"There is a turning point. In 2000, when I was still finishing my degree, I had not any political position, I entered contact with people who were concerned about these issues. Issues like the Louvain report, issues that happened in 2000 did make me open the eyes a bit ( . . . ); I took that political leap when I met people from social movements, antimilitarism, anarchist unions. And, above all, I remember the anti-globalization movement demonstrations in Barcelona and Genova. That meant a turning point for me" (quote: Alba, Zocamiñoca).*

Other interviewees pointed to specific individual experiences, for example, becoming independent from parents, which paved the way for sustaining consumption styles more consistent with their personal values, as mentioned by Brais (Zocamiñoca). Changes in residence also favored changes in food intake habits, as several interviewees indicated. This usually happens when the new locality has more options related to organic or local products which allows for experimentation with new models of consumption:

*"I lived in London before I moved to here. In London, so I did not belong to any particular cooperative, but I did go to farms that had that, this week's orders, order baskets, apart from that I was interested in all these kinds of things, because I saw a little of it. that there were, from organic markets to things a little bigger, to smaller things like this" (quote: Carlos, Zocamiñoca).*

A third turning point observed by some activists for new members to decide to join a consumer cooperative is "having a baby." This event seems to increase food concern and health awareness. However, there is no clear evidence of meaningful changes in family food habits, so as this situation is circumstantial and might not lead to the activation of environmental identity processes, as the following interviewee stresses:

*"Some people join the group because of their children. They do not consume organic food, but their children do. They come for their children. This looks a total incoherence, but they are moved by these reasons" (quote: Rocío, Semente).*

Figure 2, below, shows the concept map that relates the network of motivations underlying participation in a conscious and responsible consumption local initiative.

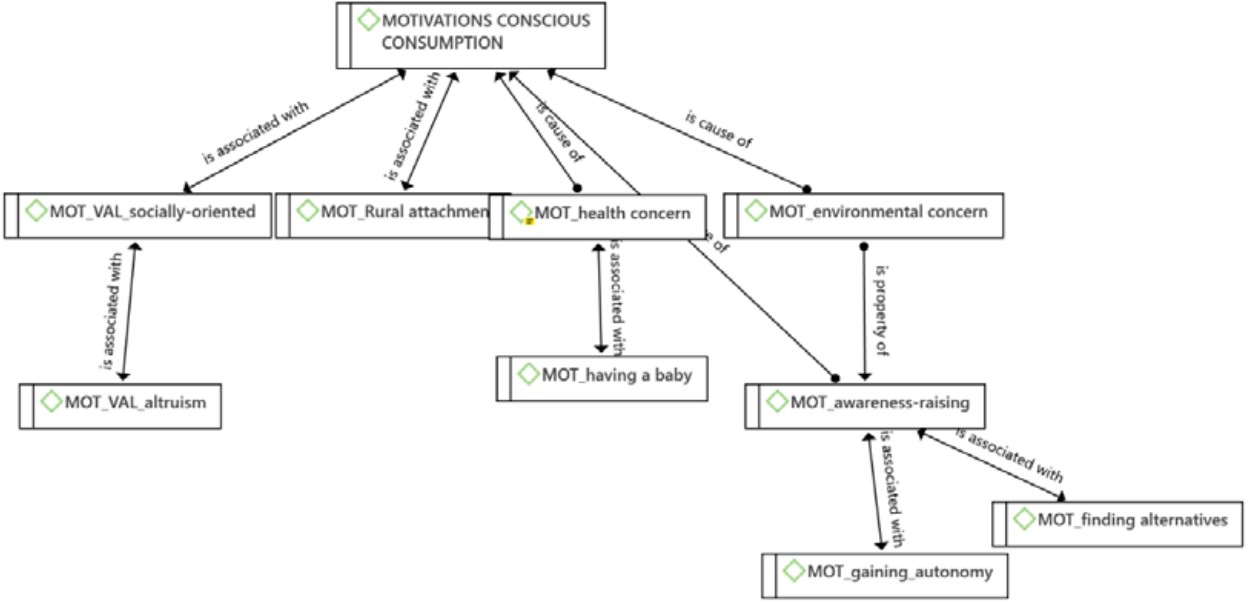

**Figure 2.** Concept map of relations between categories of motivations for conscious consumption. Source: own study (created with Atlas.ti 9).

### 4.2. Motivations for Joining a Local Conscious and Responsible Consumption Initiative

Participants in the study were asked about the reasons that prompted them to join a conscious and responsible consumption initiative. Generally speaking, most of the interviewees reported a desire to sustain a green lifestyle, perceiving CRCIs as spaces in which organic food and goods are more accessible and affordable. Participants argued motivations connected to sociopolitical activism and transformations with respect to the economy. Third, a set of motivations related to the satisfaction of psychological needs was also elicited in interviews. Table 3, below, describes these categories and indicates their relevance in terms of the groundedness and frequency in which they were mentioned in the interviews.

**Table 3.** Motivations for becoming a member of a conscious consumption initiative.

| Category | Description | Groundedness and Frequency |
|---|---|---|
| Green lifestyles | Identification of the CRCI as an enabling space for climate-friendly, fair, and ethical consumption. | G:31; F: 20 |
| Accessibility | Accessibility, availability, and affordability with respect to environmentally and socially responsible food | G: 34; F: 16 |
| Social transformation | Sociopolitical ambitions for social transformation. Desire for been involved in sociopolitical and/or socially transformative movements | G: 42; F: 13 |
| Support local economies | Support locally based economic alternatives based on the articulation of short food supply chains and foster new types of relations in economy | G: 27; F: 14 |
| New relations in economy | New relationship models in the economic context, fostering models of *prosumerism* and co-responsibility in the food system | G: 47; F: 15 |
| Self-management | Need for autonomy and control over consumption and ambitions related to self-management and gaining independence in economy | G: 36; F: 12 |
| Connectedness | Affective or relational needs: connecting with people who share common principles and values | G: 11; F: 8 |
| Competence | Individual's belief in his or her ability to accomplish a desired behavior | G: 8; F: 6 |

G = groundedness, number of citations or quotations that contain a code/category. F = frequency, number of interviews that contain a code/category. Source: own study.

### 4.2.1. Perception of Consumption Initiatives as Spaces That Facilitate Green Lifestyles

The desire to participate in a conscious consumption initiative is grounded on the perception of these organizations as spaces capable of satisfying basic food needs aligned with the participants' desire to maintain more conscious and sustainable behavior in the consumption domain. Many of the interviewees regretted that organic locally produced groceries and fair-trade products were difficult to find in conventional supermarkets, especially a decade ago, when many of these organizations emerged. Thus, as explained by one of the founding members of Aldea:

*"This cooperative was born as an attempt to make accessible in one way or another the ecological product to people" (quote: Olivia, Aldea).*

Several people interviewed in this study also referred to ethical and animal welfare concerns. The food purchased in these initiatives is considered to be more respectful of animal welfare. For example, in the case of Árbore, non-animal abuse is part of its vision and meat is not sold in the cooperative's store. Some initiatives in the network actively promote vegetarian or vegan eating styles, providing special vegan products that are difficult to reach in supermarkets.

### 4.2.2. Accessibility, Availability, and Price

The vast majority of participants interviewed mentioned, among the motivations for joining a CRCI, economic reasons (affordability), highlighting that access to organic or sustainable products through the usual marketing channels is more expensive. CRCIs provide high-quality products to their associates at more affordable prices, allowing people of different economic levels to access organic products. As a member of A Gradicela pointed out:

*"This group began with the desire to obtain organic products that there were not, except in specialized diet stores, which was very expensive" (quote: Breixo, A Gradicela).*

A new member of Semente described, in the following quotation, how the consumer group satisfied their desire to purchase baskets of fresh products that meet a series of ethical and social criteria and that were affordable for her:

*"(Q) Why did you join Semente? (A): Because it seems important to me to consume local products, to support people, farmers from here, because of the ecological footprint. And I also understand that prices are cheaper. It suits me very well. It is good. If I compare what costs what you buy in the market and in the food group, I save money, that is great" (quote: Sara, Semente).*

A few interviewees also expressed a desire for autonomy and control over their consumption choices, which becomes one of the main motivating factors for being part of a consumer initiative or even being involved in the creation of a new one. This need for autonomy is manifested as a desire to gain independence, have more information about the processes inherent to food production and distribution, and to have greater decision-making capacity and control over the products that they eventually purchase.

### 4.2.3. Sociopolitical Ambitions

Most of the participants in this study shared common ambitions related to the transformation of the current social and political system. They explain the ambition to transform the unsustainable global food system and that joining the food initiative was a suitable form to achieve this goal. One of the founding members of Árbore coop described the goals of the cooperative as:

*"A society that belongs to everyone, in which wealth must go for the benefit of society, which contrasts with capitalism, which is a form of accumulation by possession in few hands" (quote: Antón, Árbore).*

Consumption is conceived by the participants in this study as a site of political participation, "a political action" for the common good that is performed in community, *"a way of supporting the model you want, buying is a for that you can vote every day" (quote: Brais, Zocamiñoca).* However, although libertarian discourses were mentioned by the founders of several food coops, such as Árbore, Zocamiñoca, and Aldea, other interviews stated that the Galician food movement differs and is apart from radical political positions such as libertarian or anarchism discourses, which did inspire the constitution of similar movements in other parts of Spain.

### 4.2.4. Fostering Local Economies and New Relations in Economy

In line with the above, many of the participants argued a desire to foster local economies by supporting locally based economic alternatives that prioritize the social wellbeing of the community, such as of the social and solidarity economy movement. Thus, Galician food activists aimed to build new types of relations in the economy, developing models of prosumerism and co-responsibility in the food system. Some of the respondents stressed the need to build new types of relations between consumers and producers based on mutual respect, proximity, and empathy, recognizing and dignifying the work of "small farmers" in contrast to the current relations based on struggling organic and traditional forms of agriculture.

Participants also expressed the desire to join a community-based project that has the potential of transforming utopia into reality. Food projects are perceived as "something tangible, capable of reaching an impact, in the local context", as stressed also by the following participant:

*"I joined the group with the idea of changing the world, "making the revolution" one could say ( . . . ) My basic motivation is political. I needed to participate in social groups that tried to transform the reality we live in. That was why I joined the group. Trying to engage in collective work" (quote: Victor, Agrelar).*

### 4.2.5. Self-Management and Autonomy

The category self-management relates to the need for gaining autonomy and control over consumption by creating and being part of independent organizations that challenge conventional market structures. For instance, Galician food activists argued that collaborative models are more effective and achieve greater transformative capacity than the more individualistic formulas, such as a boycotting. Grassroots movements are perceived as more effective in promoting structural changes, democratizing the economy:

*"From an economic point of view, we must set up structures that are ours. In which we decide. It cannot be that democracy is political, but in the economic part, the economy*

*is in the hands of few guys who decide everything. Because if the citizens do not decide in the economy, what kind of democracy is that? And if there are projects like this, or different, that is where you decide in the economy" (quote: Antón, Arbore).*

Collective forms of consumption rely on a preference for self-managed or cooperative models that are governed by the principles of equality in decision-making organizations *"where there are no power relations or hierarchies" (quote: Fabio, Zocamiñoca)*, but models of horizontal participation that favor symmetrical relationships, as described below:

*"In my case, I am aligned with the self-management philosophy of these groups. I was in the scouts Before, which was a very self-managed group, but very pyramidal. Food coops are not hierarchical structures, which interests me the most. That the tomatoes are organic is fine, it is important, but that (self-management) depends on each group and each person ( ... ). The step of entering these (consumer) groups was more related to looking for another type of participation, different from the one I had experienced, not so hierarchical" (quote: Alba, Zocamiñoca).*

4.2.6. Satisfaction of Affective or Relational Needs: Connectedness with Like-Minded People

Several interviewees manifested a desire to relate or connect with other people with whom they share similar values and goals and concerns. For example, one of the members of Agrelar recognized that one of the main reasons to join the group concerns the desire to relate with like-minded people with whom to develop transformative projects in the field:

*"I needed to participate in some social group that tried to transform the social reality we live from. That was why I joined. To try to do collective work. Here in the town, I felt alone (laughs), trying to do with own garden but without ... I was not even capable of producing for my own consumption, or of doing anything else. So that would be the main motivation to join the group" (quote: Victor, Agrelar)*

This need for connectedness is expressed as a desire to engage in communities of practice that share knowledge and mutual support to their members and allow for personal and collective transformation. This motivation was reported particularly by the founding members of Zocamiñoca cooperative, as illustrated in the following quotes:

*"This is a project to transform ourselves as well. We want the people who enter here to learn to function differently" (quote: Xulia, Zocamiñoca)*

*"It is an opportunity to explore or promote other things that you do not like about society in general and say, because we have the opportunity to be consistent and to try to work differently. This relates to the culture of the group. The issue of allowing us to disagree, allowing doubts to be welcomed" (quote: Gael, Zocamiñoca)*

Finally, it can be concluded that people involved in the Galician consumption initiatives express a wide range of motivations that combine health concerns and environmental awareness, all of which lead to the desire of perform green lifestyles. Third, sociopolitical ambitions are also involved, as activists endorse transformative goals aligned with the social and solidarity economy movement, supporting local economies, and being capable of maintaining traditional lifestyles in rural areas, as described as follows:

*"It has to do with conscious consumption. On the one hand, if there are fair trade products, I know that labor rights are respected. We can trace it. On the other hand, eating organic food means that nature is respected, cared, the principles of organic farming are respected, as opposed to industrial agriculture. I intake products that are better for my health" (quote: Tomás, Árbore)*

*"We are talking about three spheres here. First, the personal sphere, in terms of the type of food I want to eat; There is the social level of what kind of employment I want to promote. And third, there is the local sphere, what economic model do I promote and want to support" (quote: Brais, Zocamiñoca)*

These three dimensions are reflected in several interviews and, based on these results, it seems to be the convergence of these three types of motivations that would lead a person to make the decision to join a conscious and responsible consumption initiative; if only one or two of these motivations are present, they could be satisfied by conventional establishments or local markets. Figure 3, below, illustrates the conceptual map of motivations for conscious consumption in the frame of the consumer initiatives and the types of relations between the distinct categories identified in the study.

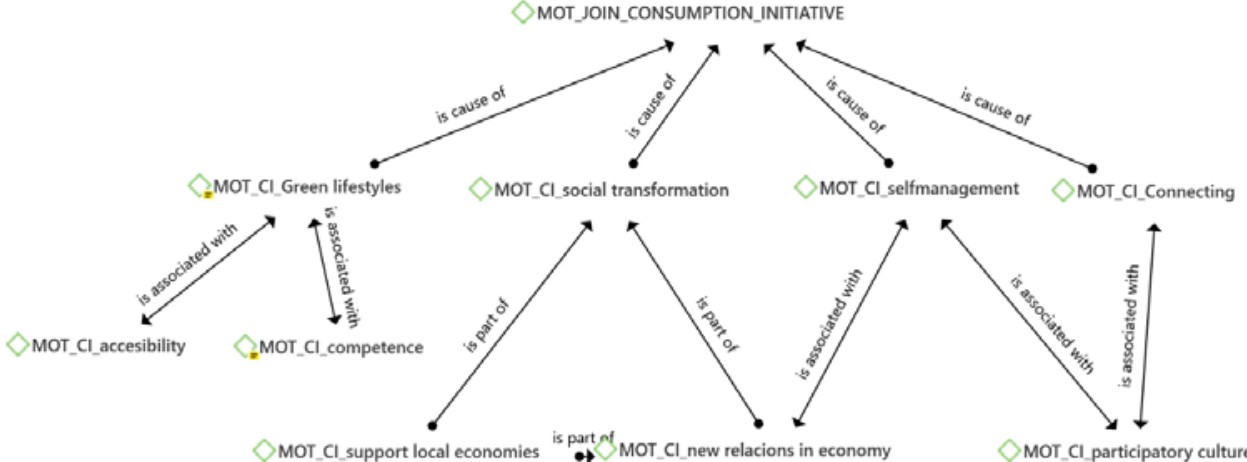

**Figure 3.** Concept map of relations between categories of motivations underlying participation in a conscious and responsible consumption initiative (CRCI). Source: own study (created with Atlas.ti.9).

## 5. Discussion and Conclusions

This study contributes to the understanding of the motivations and aspirations underlying sustainable consumption. The empirical research was conducted in a selection of citizen-led social innovations in the food domain, involving significant sample of members of local conscious and responsible consumption initiatives who, in their daily lives, dedicate their money, time, and efforts to sustaining these alternative consumption spaces. Sustainable consumers' motivations have been explored in relation to the engagement or construction of alternative modes of social organization in the food system, which has become an innovative approach for psychological and social innovation studies [8,9,37]. In agreement with the findings of previous studies on organic food consumption [35], three types of motivations for individual's sustainable consumption can be distinguished: (i) self-oriented motivations that serve individuals' wellbeing; (ii) socioenvironmental consciousness that relates to the individual's environmental identity and personal values; and (iii) sociopolitical motives that connect to a strong discontent with the food economy systems and the ambition to change current power relations in society.

Concerning the self-oriented motivations, the findings of this study emphasize the attribution of specific features to organic products (e.g., superior quality and healthiness). Thus, consumers see organic groceries as a means to sustain healthy diets due to the high quality attributed to seasonal and organically produced food. This confirms previous studies that suggest that healthiness, superior quality, and good taste are the specific attributes that drive organic food consumption [21–24]. Concerns about the health risk from food consumption is central in the reasoning of the consumers, while hedonistic goals were only reported occasionally. This contrasts with other works that found that the experience of enjoying a better tasting product is an attractive feature for organic products buyers [45] or Slow Food activists [37].

Galician sustainable consumers shared prominent levels of environmental literacy and socioenvironmental consciousness regarding the ecological crisis and the contribution of the food system to nature deprivation and climate change. This feature was found to drive the purchase of organic groceries and other environmentally friendly manufacturing goods.

Concurrently, they usually support individuals and institutions taking actions towards environmental protection [45], and they often declare behaviors supporting sustainability, such as saving resources, recycling, or low carbon transportation, albeit with different levels of commitment.

The participants in this study do experience a strong sense of responsibility towards the environment, which has a positive and direct impact on their actual purchase behavior. They perceive themselves as part of the solution, feeling responsible for acting and fostering a change toward more global sustainability, as found in previous studies [36,45,46]. Such individual responsibility connects, then, with sociopolitical aspirations that are manifested through the aim of challenging dominant power structures and building alternative fair, sustainable, and democratic practices in the economy, e.g., by supporting local farms or strengthening short food supply chains [9,35,47]. In their reasoning, their discourse also connects to their ambition of contributing to the conservation of Galician rural areas, to which they feel emotionally attached. This is consistent with the research on place attachment that has examined people's feelings of connection to specific physical environments, including natural environments. Thus, many studies found that awareness of the environmental, social, and economic impact of consumption is a direct antecedent for responsible behavior [48,49].

The findings of this study point to the interlinkages of both self-oriented motivations and sociopolitical ambitions as the basis for citizen's engagement in conscious and responsible consumption initiatives. First, most of the respondents identified conscious and responsible consumption initiatives as suitable spaces able to fulfill their basic needs related to food intake and wellbeing in a more available and affordable way. As previous studies indicated, higher prices and the lack of economic resources are major barriers to purchasing sustainable products, together with the limited availability of organic food [13,22,23]. Conversely, these initiatives often facilitate access to organic and ethically produced products at lower prices than supermarkets and organic shops, although they usually require affiliation or a certain degree of commitment.

Consistent with previous research [14,37,47], sociopolitical ambitions were also primary motivations for consumers to engage in collective models of food consumption. These local initiatives are perceived as manifestations of new social movements that attempt to challenge the unsustainable practices that characterize the dominant food system. Participants stress their ambition to change the relationships between consumers and producers, fostering mutual respect, proximity, and empathy, recognizing and dignifying the work of the farmer or producer. The satisfaction of political goals is thus experienced through the construction of shared spaces, socially innovative practices, and discourses aiming for sustainable and democratic structures in the economy [30].

The results show that sustainable lifestyle initiatives have a positive impact in terms of the satisfaction of basic psychological needs and thus support the achievement of higher levels of autonomy, competence, and connectedness. Several interviewees report a desire for autonomy and control over their purchasing decisions. The "desire to feel free to consume the way one would like to" led them to join a local consumption initiative [50]. Further, a few participants in this study report being motivated by the desire to socialize with like-minded people and to engage in community-led projects led by people that share a common vision, values, goals, and concerns, which eventually satisfies the need for belonging. These findings are in agreement with previous studies that explored empowerment and agency processes in transformative social innovations in which practitioners were found to be empowered by both the satisfaction of intrinsic motivations and the achievement of sociopolitical goals [30,32,34,35].

In conclusion, the decision to join a consumer's initiative appears to be the result of the cognitive process of reflection on the best mode of satisfying intrinsic basic needs (e.g., wellbeing, need for connectedness, or autonomy), achieving certain sociopolitical ambitions and consuming in accordance with their environmental values. However, if only one or two of these factors were predominant, individuals would satisfy their needs and aspirations

in other types of providers (e.g., supermarkets or organic shops). This becomes a relevant input for those initiatives which aim to attract a wider range of associates or consumers. For example, healthiness attributes and superior food quality may result in effective arguments to attract new members, addressing the satisfaction of their personal needs (e.g., health) and their sense of environmental and social responsibility, which may further increase the consistency between attitude and actual purchasing behavior [10]. More emphasis could also be placed on personal aspirations related to the fulfilment of psychological needs and intrinsic motivations, as a better sense of community and a desire for more meaningful relationships have been stressed as efficient approaches to foster green lifestyles [50]. Food co-ops could develop targeted marketing strategies, appealing to the environmental values of the audience but also stressing the transformative potential of collective consumption.

Further, alternative approaches may focus on the benefits of collective consumption, stressing the idea that conscious consumers are not just individuals but part of a local or global community that seeks alternative food systems that protect both the individual and environmental wellness. One interesting finding of this study relates to the occurrence of specific "life events" that appear to function as triggers for sustainable lifestyles. Participation in social movements, changes of residence, or the birth of a child encouraged people to experiment with new dietary options or be more willing to purchase organic and healthy products. All of them are potential opportunities to approach new members by targeted messages addressing specific needs and expectations.

- Policy and practical implications

This study has significant practical implications. Scientific evidence from this and previous studies suggests that consumer behavior is not only affected by attitude, but also by various other personal and situational factors. Therefore, authorities should focus their attention on challenging important situational conditions (e.g., price or the lack of accessibility) that impede the wide adoption of low carbon consumption practices. Moreover, sustainable consumption "demands more sophisticated policy approaches" aimed at making desirable climate friendly behaviors easy ([51], p. 262). For instance, institutions can find in grassroots social innovations in the field powerful allies that may play a key role in transitions towards green economies at the local and regional level by fostering new pathways in sustainable agriculture and support innovative policies. As environmental awareness and social values are key factors underlying sustainable consumption, new environmentally friendly practices should be promoted by environmental education policies raising knowledge and awareness of the climate impact of food production and consumption practices as well as the best strategies to tackle climate adaptation [13]. This will lead in increasing consumers' knowledge about their impact on food systems, and especially about how their collective behavior supports alternative agriculture and rural lifestyles.

The results of this study have interesting implications for the marketing of organic products. Food suppliers could be much better at servicing households, providing effective information and support about healthy diets. Targeted information about food safety and the organic production and processing increases the level of trust in sustainable goods. The functional characteristics of organic food combined with high quality products positively influence consumers' green purchase behavior. Thus, healthy properties should be coupled with references to other values such as hedonism, pleasure, or sustainable (local) development, which also act as stimuli for sustainable consumption, as previous studies in social marketing have pointed out [32]. The findings of this and previous studies suggest that shopping can become a pleasurable experience that produces a pleasant psychosocial benefit in consumers [32], increasing their willingness to be a member of food co-ops or strengthening bondsbetween with their members clients. This understanding will enable organic shops—as well as conscious and responsible consumption initiatives—to formulate marketing strategies to encourage conscious consumers to buy within or to join their initiative.

- Study limitations and future research

As in all other studies, there are limitations to be mentioned. First, a convenient sample of participants was used for this study, which does not cover the substantial number of local manifestations of collective consumption existing across the Galician territory. This issue was dealt with by enlarging the number of cases involved in the second round of interviews, as well as including different forms of organization which increased the diversity of the sample. Furthermore, recruitment consisted of an invitation open to the food initiatives and only a limited number of members positively responded the call.

Despite gathering a representative sample of the population involved in the case study was not the intention of this study, it would be useful to adopt, in future research, a purposive sample strategy that involves a range of sociodemographic characteristics (including gender and age), along with a balanced sample of participants representing distinct levels of engagement in conscious consumption. This would allow researchers to obtain a more complete picture of the different typologies of sustainable consumers, which would be the basis for targeted policy interventions or social marketing strategies [35,45].

Future research could also employ different methodological strategies to understand the psychological, cognitive, and emotional processes involved in consumer choices [30], or to measure the influence of specific motives and dynamics that might drive or inhibit citizen engagement in collective forms of consumption.

**Author Contributions:** Conceptualization, I.L.-B. and R.G.-M.; Methodology, I.L.-B. and R.G.-M.; Formal analysis, I.L.-B.; Investigation, I.L.-B.; Data curation, I.L.-B.; writing—original draft preparation, I.L.-B.; writing—review and editing, R.G.-M. and J.-M.M.-C.; supervision, R.G.-M. and J.-M.M.-C. All authors have read and agreed to the published version of the manuscript.

**Funding:** This research received no external funding.

**Informed Consent Statement:** Informed consent was obtained from all participants involved in the study.

**Data Availability Statement:** Not applicable.

**Acknowledgments:** The authors would like to acknowledge the support provided by the local initiatives of the Galician Conscious and Consumption Network and, specifically, to the interviewees that participated in this study.

**Conflicts of Interest:** The authors declare no conflict of interest.

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
