# Peer review of "Understanding Motivations for Individual and Collective Sustainable Food Consumption: A Case Study of the Galician Conscious and Responsible Consumption Network"

_sustainability, doi:10.3390/su15054111_

Round 1

Reviewer 1 Report

The article "Understanding motivations for individual and collective sustainable food consumption. A case study of the Galician Conscious and Responsible Consumption Network" addresses an interesting topic. However, in its current form the article does not provide any new nor enriching insights into the motivations of consumers. Again, the combination between consumer motivations, buying behaviour and sustainability could be quite interesting. As a matter of principle, buying behaviour is based on the consumer’s belief, a product with its different attributes is a mean to an end. This perception is reflected in Means-End-Chains-Theory, which implies that product attributes are of little importance to the consumers who try to achieve benefits, i.e. certain consequences, with the aid of product attributes. Hence, product attributes only have a meaning in terms of the consequences they bring about. However, the prospect of buying different products is not to achieve consequences. In fact, consequences in turn are perceived to satisfy higher needs, i.e. personal values which are defined as enduring beliefs that specific modes of conduct or end-states of existence are personally or socially preferable to opposite modes of conduct or end-states of existence (OLSON and REYNOLDS, 2001: 13). Therefore, the expectation of achieving a personal value through the usage of a certain product is the actual driving force of observable consumption patterns (GRUNERT, 1994: 218; GUTMAN and REYNOLDS, 1979: 132). To analyse the connection between product attributes, consequences and a consumer’s personal values, i.e. a consumer’s Means-End-Chains, consumer research uses laddering-interviews. Whereas, results are presented using a graphic representation of the associations made by the consumers called Hierarchical Value Map. All-in-all, using the "Galician Conscious and Responsible Consumption Network" as the base for a study based on "Means-End-Chain" would be highly recommendable.

Author Response

Thank you very much for your review and comments to the first version of our paper. We took your comments and suggestions into consideration  and, coherently, changes have been made on the revised version of the paper, as we explained, in a more detailed way, in the attached document. 

Reviewer 2 Report

This article focuses on the individual motivations that drive conscious consumption in both individual and collective spheres from Spain (in Galacian region).

The literature review is rich and varied based on sustainable consumption.

Figure 1 doesn’t contribute to the article.

The results are very interesting and contribute to new insights.

I give this article a favorable opinion.

Author Response

Thank you very much for your review. We took into consideration your comment concerning the relevance of Figure 1. We decided to substitute this figure by a new one that illustrates the distribution of the different conscious and responsible consumption initiative across the Galician territory (see revised version of the paper: Figure 1. Collaborative map of the Galician Consumption Network, p.3). We hope this helps to the better understanding of the case study and the relevance of our case study. 

Reviewer 3 Report

Dear Authors

The issues of consumers’ behaviour and sustainability consumption has been my scientific interest for years. That is why I am very happy to have the opportunity to read your very interesting article on this topic.

The manuscript entitled “Understanding motivations for individual and collective sustainable food consumption. A case study of the Galician Conscious and Responsible Consumption Network” is well-written and has a research character. The Authors should be appreciated for the research reliability and methods used. The strong points of this article are also its layout and the clarity of presented contents.

The introduction should be an introduction, not a review of the literature. The literature review should be presented in a separate section.

The authors should have put forward hypotheses or at least research questions. Why didn't they?

The references are impressive and closely related to the topic of the article. Taking advantage of the fact that both Authors and me are passionate about the issues of consumers’ behaviour and  consumption, I would like to draw their attention to two works that can possibly enrich the discussion of results. Here they are:

Maciejewski G., Malinowska M., Kucharska B., Kucia M., Kolny B. Sustainable development as a factor differentiating consumer behavior. The case of Poland, European Research Studies Journal 2021, 24(3), 934-948. DOI: 10.35808/ersj/2392

Wang, J., Yu, X. The Driving Path of Customer Sustainable Consumption Behaviors in the Context of the Sharing Economy - Based on the Interaction Effect of Customer Signal, Service Provider Signal, and Platform Signal. Sustainability 2021, 13(7), 3826, 1-16, https://doi.org/10.3390/su13073826.

In the end, just a little thing. The authors place tables and figures in the text. However, the source of the information is nowhere to be found. If they were created as a result of own work, please mark it, for example: "Source: own study".

I hope that the indicated remarks will help the Authors to improve their text even more so that the work will be published. Good luck!

Author Response

Thank you very much for your review and comments to the first version of our paper. We took all your comments into consideration  and, coherently, changes have been made on the revised version of the paper, as we explain. in a more detailed way in the attached document. 

Round 2

Reviewer 1 Report

The revised version has taken up some of the remarks. Still the overall empirical part could be improved. However, as it is not realistic to re-do the whole empirical part, the revision is ok.